

# Integrating micromagnets and hybrid nanowires
# for topological quantum computing

Malcolm J. A. Jardine[1], John P. T. Stenger[1], Yifan Jiang[1], Eline J. de Jong[2],
Wenbo Wang[3], Ania C. Bleszynski Jayich[3] and Sergey M. Frolov[1]

**1** Department of Physics and Astronomy, University of Pittsburgh, Pittsburgh, PA, 15260, USA
**2** Department of Applied Physics, Eindhoven University of Technology,
5600 MB Eindhoven, The Netherlands
**3** Department of Physics, University of California, Santa Barbara CA 93106, USA

## Abstract

Majorana zero modes are expected to arise in semiconductor-superconductor hybrid systems, with potential topological quantum computing applications. One limitation of this approach is the need for a relatively high external magnetic field that should also change direction at the nanoscale. This proposal considers devices that incorporate micromagnets to address this challenge. We perform numerical simulations of stray magnetic fields from different micromagnet configurations, which are then used to solve for Majorana wavefunctions. Several devices are proposed, starting with the basic four-magnet design to align magnetic field with the nanowire and scaling up to nanowire T-junctions. The feasibility of the approach is assessed by performing magnetic imaging of prototype patterns.

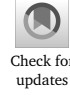

# 1   Introduction

One-dimensional topological superconductors host Majorana zero modes (MZMs) [1,2]. They are promising for fault-tolerant quantum computing because of the predicted topological protection that they facilitate [3,4]). While there has been considerable effort applied to identify MZMs in semiconductor nanowires [5–7], the evidence to date is not conclusive due to plausible alternative explanations such as non-topological Andreev Bound States [8,9]. With new developments in materials research and experimental methods it is reasonable to expect that the present day challenges can be overcome [10]. We look beyond to explore how generating local magnetic fields using micromagnets can aid in the design of Majorana devices.

There have been several studies on incorporating magnetic materials to induce MZM in hybrid systems, though the questions asked were different. One class of ideas has focused on generating synthetic spin-orbit coupling in weak spin-orbit materials through nanomagnet patterning [11–18]. Another class imagines shells of magnetic insulators on nanowires as a path to topological superconductivity through using exchange interactions [19, 20]. These results have sparked a debate about the feasibility and true origin of the observed signals [21–24].

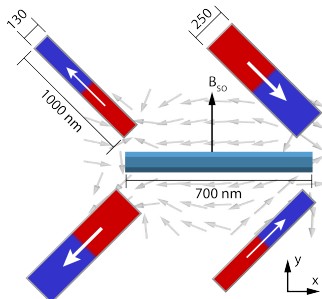

Figure 1: The Dragonfly setup with four micromagnets (blue/red) and an overlay of the magnetic field calculated with MuMax3 (gray arrows). The nanowire runs horizontally with the spin-orbit axis vertical, indicated by $B_{so}$.

In this work we propose how stray magnetic fields can be used to realize basic two-Majorana building blocks, as well as four-Majorana fusion and six-Majorana T-junction braiding devices. Locally inducing magnetic fields avoids the reliance on a global magnetic field, and therefore fields can be oriented differently in the nanowire set-up opening up possibilities in measuring of complicated structures like T-junctions braiding devices and the ability to address and manipulate individual MZMs. In the basic building block (Dragonfly, Fig. 1), four micromagnets are arranged around the nanowire such that field lines flow along the wire for 700 nm. We find through modular micromagnetics and Schrödinger calculations that it should be possible to enter the topologically superconducting state and achieve partial Majorana separation. Higher stray fields, e.g. from stronger or thicker magnets, would make the regime more robust. We perform magnetic force microscopy on prototype micromagnet patterns and

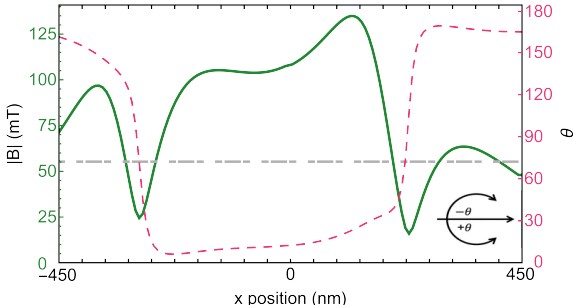

Figure 2: Magnetic field profile, as a function of position, of the four magnet Dragonfly setup. Solid line is field amplitude and dashed line is field angle $\theta$ relative to negative x-axis (inset). The field is averaged over a hexagonal cross-section of the nanowire with a characteristic dimension of 100 nm. In this Figure only, the reader is invited to imagine an infinitely long nanowire without ends and consider what field profile would be created along such a nanowire. The horizontal dashed gray line indicates a uniform field for entering the topological regime in an infinite nanowire.

find that it is possible to realize the building block Dragonfly configurations, though arranging all micromagnets in a T-junction to the desired orientations will require very accurate control of switching fields.

## 2 Numerical Results

**Brief Methods** The micromagnetic simulation software MuMax3 [25] is used to simulate realistic single domain-sized micromagnets, including hysteretic magnetization and stray fields. Hysteresis simulations are performed to obtain the required magnetization state. The parameters for cobalt were used, details can be found in section A.4. The stray magnetic fields are integrated over an imagined hexagonal nanowire cross-section to obtain a one-dimensional field profile which can in principle extend to infinity. The field profile is used as input into a basic one-dimensional Majorana nanowire model [26] to obtain the energy spectrum and calculate Majorana wavefunctions $\gamma_1$ and $\gamma_2$. For this step a finite nanowire length is chosen over which the field profile can be utilized for a given set-up.

**Dragonfly setup** The setup presented in Fig. 1 is our basic configuration. The four micromagnets, that resemble a dragonfly, induce field lines along the nanowire, coming in at the right end, and flowing out of the left end. The field does not deviate by more than 30° over a 700 nm long nanowire segment, see Fig. 2, and maintains a relatively uniform amplitude. Furthermore, for a given hexagonal cross-section the standard deviation is of order 5% which justifies the procedure of producing a one-dimensional field profile by averaging over the cross-section, at least for the central segment of the nanowire length, see Fig. 17. The nearly parallel field is required for MZMs formation, since such field is mostly orthogonal to the effective spin-orbit field, $B_{so}$ [1,2,27,28], hence why we use angle relative to the nanowire when presenting the stray field profile. Four, rather than two, magnets are required to cancel y-fields and enhance x-fields, with no external field required. Micromagnets placed parallel to the nanowire, e.g. as a shell around the nanowire, produce largely y-fields that are highly non-uniform, concentrated outside the poles of the magnets (see Fig. 15 [19]).

The micromagnet widths are different to ensure different switching fields and to aid in the preparation of a desired mutual magnetization pattern [29]. The magnets are rotated at 45

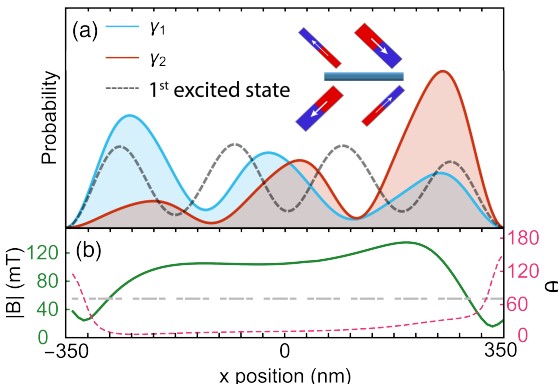

Figure 3: (a) Probability distributions for two Majorana wavefunctions $\gamma_1$ and $\gamma_2$. The first excited state (dashed line) is a bulk nanowire state. (b) Magnetic field profile reproduced from Fig. 2 over a smaller range. Hard boundaries are introduced at ±350 nm to calculate the wavefunctions. There is no applied external field.

degrees to aid magnetization through the hysteresis simulation. Magnets on opposite sides of the nanowire are perpendicular, and therefore the external field used for magnetization is also perpendicular for the pairs of magnets. The out of plane direction (z-component) of the simulated field is insignificant due to the symmetry of the micromagnet arrangement (see supplementary information 13).

Given the simulated magnetic field profile along the 700 nm long nanowire, there is clear Majorana polarization of the two MZMs, $\gamma_1$ and $\gamma_2$, shown in Fig. 3. Though we assume a relatively low magnetic field for the topological transition, 50 mT. The two Majorana wavefunctions are highly overlapping. The lowest (partially separated Majorana) and the first excited energy levels have energies $E_0/\Delta = 0.28$ and $E_1/\Delta = 0.82$, where $\Delta$ is the superconducting gap. We note that a small global external field can be applied to boost Zeeman energy.

**Double Dragonfly** The next device (inset Fig. 4) combines two Dragonfly set-ups and an additional vertical magnet (dashed border), elongating the topological nanowire region. The concept can be repeated multiple times to further extend Majorana separation or create multiple pairs of MZM, for instance in Majorana fusion experiments [4]. The right Dragonfly unit is reflected: this means magnetic fields point opposite in the left and right nanowire segments. Naively, MZM generated by opposite magnetic fields cannot couple as they possess opposite spin [30]. Nevertheless, left and right MZM form a pair due to field rotation provided by the central magnet. Using the same topological criterion we now find a more substantial MZM polarization of the ground state $E_0$ (red/blue) with more noticeable tailing off of the wavefunctions. This is because the total length of the device is increased and two MZMs in the middle (dashed line in Fig. 4a) hybridize and reduce their overlap with end MZMs. Application of an external field (10's of mT) in the positive y-direction in this case aligns the total field closer with the nanowire. Supplementary information presents other versions of the double Dragonfly.

**T-junction braiding device** The T-junction set-up, shown in Fig. 5, explores how micromagnets could be used to realize a MZM braiding setup. The horizontal left and right arm sections are made by chaining together Dragonflies, and a perpendicular leg section in the middle has a micromagnetic configuration similar to a single Dragonfly. Adding a second dragonfly to the leg does not substantially improve MZM separation due to an external field in the y-direction

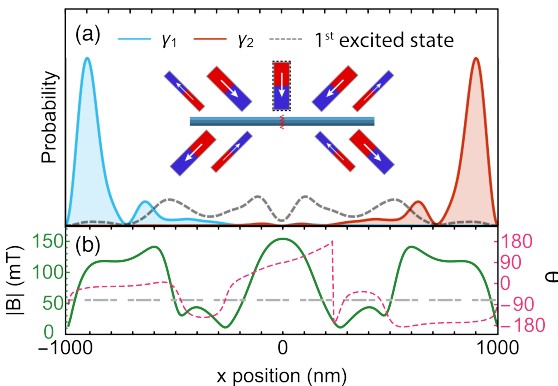

Figure 4: (a) Probability distributions of two lowest energy states for double Dragon-fly setup with gate open (shown in inset). MZM (red/blue) with $E_0/\Delta = 5.1 \times 10^{-4}$. The first excited state (grey dashed) with $E_1/\Delta = 1.6 \times 10^{-1}$. External field of 40mT is applied in positive y-direction. Inset: two dragonfly configurations, with zigzag indicating an electrostatic gate capable of dividing the nanowire in two parts. (b) Magnetic field profile along the wire, showing amplitude (solid) and angle (dashed).

(see supplementary information). This allows the field to be parallel to the nanowire in perpendicular sections of the wire, something that is not possible in a uniform external field.

The simulated stray field (supplementary Fig. 14a), with an additional y-direction external magnetic field of 40 mT, allowed MZMs to separate or couple across any two sections of the T-junction nanowire, seen in Fig. 6, where each arm of the junction has an electrostatic gate, $G_1$, $G_2$ and $G_3$, to control which sections are connected and disconnected.

With all three gates pinching-off the nanowire we see that the three lowest energy ground states are pairs of MZMs (red, green and blue) confined to separate wire segments (Fig. 6a). With gate $G_3$ cutting off the perpendicular leg section, Fig. 6b, the lowest energy state (red) is localized at the ends of the top wire, demonstrating that the micromagnets allow this to act as a single, long topological superconductor. The second lowest energy state of the entire system is MZMs isolated in the perpendicular nanowire (blue), and the third lowest state (green) has weight mostly at the junction, representing two nearby MZM. We note that the fourth energy state is larger in energy with more weight in the bulk of the nanowire, suggesting it is not a partially hybridized MZM pair.

By pinching off gates $G_1$ or $G_2$, Figs. 6c and 6d, we cut off either the left or the right section of nanowire. MZM states couple and decouple across the T-junction, In panel (d) the blue wavefunction extends over entire left and leg segments of the T-junction, making the red wavefunction appear purple. These results suggest that a repeated Dragonfly setup could be used to realize a braiding setup, in principle, although with relatively complex micromagnet configurations.

## 3 Magnetic Force Microscopy

We take the first step to evaluate these device concepts experimentally, by studying magnetization patterns of micromagnets. Ferromagnetic micro-strips are fabricated in a design that represents a simplified version of the braiding T-junction setup (Fig. 7). The design features three Dragonflies arranged in a T with the same dimensions as Fig. 1 but a shorter distance between magnets, and an extra vertical magnet. The strips are written by electron beam

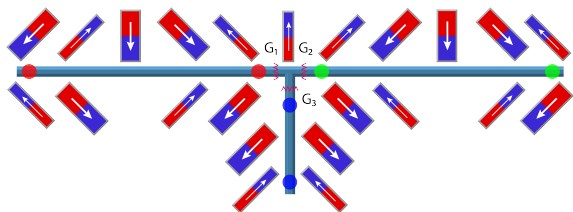

Figure 5: T-Junction setup. The top two sections of nanowire are 5000 nm in length
in total and the perpendicular section is 1100 nm. Zigzag lines indicate electrostatic
gates. Circles indicate desired positions of 6 MZM with all gates on.

lithography (EBL) and the metal is deposited by electron beam evaporation from a CoFeB
source (atomic ratio 30/55/15 before deposition) to a thickness of 20 nm. This is thinner
than typical nanowires, and was done to reduce magnetic signal and permit higher resolution
imaging. It is believed that the final strips are mostly CoFe without boron [31].

Magnetic force microscopy (MFM) and atomic force microscopy (AFM) are performed on
triple Dragonflies (Fig. 7). An attempt is made to take advantage of hysteretic magnetiza-
tion and prepare magnets preferentially in the desired Dragonfly configuration so that stray
magnetic fields are along the imaginary nanowire in between the micromagnets. The range
of field where wide and narrow micromagnets are antiparallel is determined from separate
SQUID measurements to be between 15 and 20 mT, in agreement with Ref. [29], when re-
duced CoFe thickness is accounted for.

Twenty-four T-junctions with three Dragonflies each are imaged, and six of 72 total Drag-
onflies are magnetized as required for MZM generation. Fig. 7(b) shows one such section with
the magnetization marked by arrows (supplementary Fig. 18 shows all data). The occurrence
of four micromagnets in the right orientation is consistent with random magnetization. While
in our MuMax3 simulations it is possible to run through the hysteretic magnetization cycle
and prepare micromagnets in the desired configuration, experimental variations in switching
fields highlight a challenge. Optimized micromagnet fabrication will yield sharper switching
and more reproducible switching fields in the future.

## 4 Conclusions, limitations

We consider device concepts in which micromagnets generate stray field patterns suitable for
the generation of Majorana zero modes. Our approach assumes micromagnets placed next to
semiconductor nanowires that possess strong spin-orbit coupling, and are coated with super-
conducting shells. The requirements on the stray magnetic fields are that they are of sufficient
strength to drive a topological transition, and should be oriented as much as possible along
the nanowire. The building block of our magnetic design is a Dragonfly configuration in which
four micromagnets are magnetized such that the magnetic field lines flow out of one pair of
micromagnets, along the nanowire, and into the other pair (Fig. 1). By repeating the Drag-
onfly pattern along the nanowire, we can extend the length of the topological segment with
addition of coupling magnets. The approach can also be applied to T-junctions required for
Majorana braiding experiments, in which case magnetic field turns into the T-junction leg that
is perpendicular to the junction top.

Among the limitations is the still limited strength of stray fields from micromagnets. In
previous experiments the typical stray fields are in the range of tens of milliTesla [29]. This is
in principle sufficient to enter the topological regime in large g-factor semiconductor such as

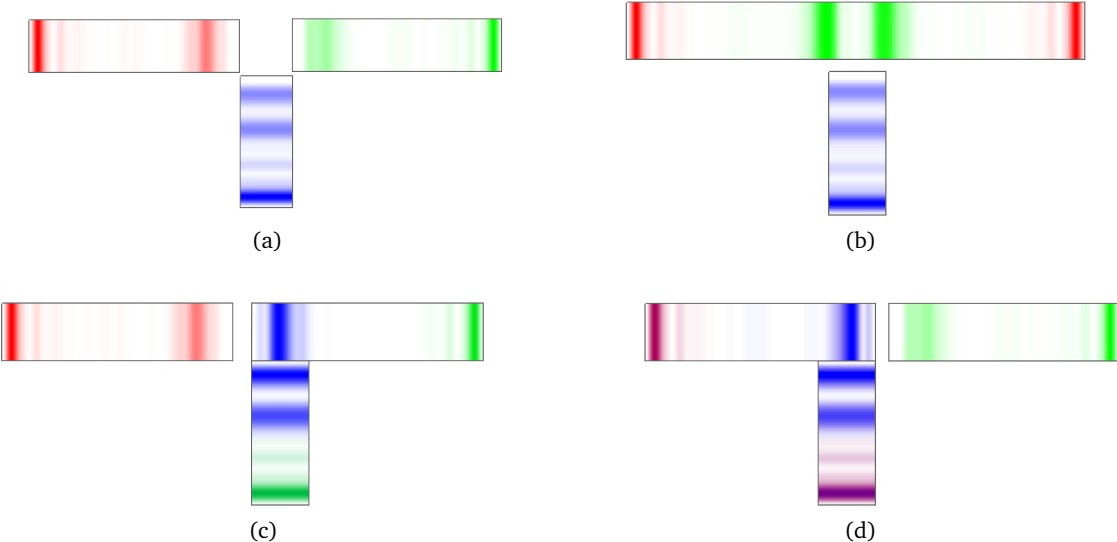

Figure 6: Probability distributions (color) of the ground, second and third lowest energy with (a) all gates (b) gate $G_3$, (b) gate $G_1$, (c) gate $G_2$ activated. Colors chosen so that red wavefunction always has a weight on the left end. The energies for the pairs of Majorana states are given in the table below.

| Energies of setup $/\Delta$ | (a) | (b) | (c) | (d) |
|---|---|---|---|---|
| $E_0$ | $1.6 \times 10^{-3}$ | $1.6 \times 10^{-5}$ | $3.5 \times 10^{-4}$ | $7.2 \times 10^{-4}$ |
| $E_1$ | $2.3 \times 10^{-3}$ | $9.4 \times 10^{-3}$ | $1.6 \times 10^{-3}$ | $2.3 \times 10^{-3}$ |
| $E_2$ | $9.4 \times 10^{-3}$ | $4.2 \times 10^{-2}$ | $4.3 \times 10^{-3}$ | $3.1 \times 10^{-3}$ |
| $E_3$ (not shown) | $1.5 \times 10^{-1}$ | $1.1 \times 10^{-1}$ | $1.3 \times 10^{-1}$ | $1.2 \times 10^{-1}$ |

InSb nanowires, but it limits the parameter space for Majorana separation and manipulation. Stronger magnetic materials, or thicker micromagnets can help. Additionally, the Meissner effect from the superconducting layer on top of the nanowires was not taken into account, however the effect on the magnetic field is assumed to be small as magnetic fields penetrate through such thin superconducting films. Disorder in real nanowire devices will put limitations on the achievable MZM separation distance.

A challenge that became apparent from magnetic imaging of prototype structures is how to prepare all micromagnets in the appropriate relative magnetic orientation. This becomes harder when the configurations increase in complexity such as for T-junctions. Though basic two-Majorana experiments should be possible already now, in the future better control over coercive fields, further pattern optimization, and direct magnetic writing can be deployed.

Among future ideas, a promising path is using a Y-junction instead of a T-junction [32], an approach that may require fewer micromagnets as some can be shared between the branches. Rather than gate-controlled MZM coupling, magnetic field mediated coupling can be investigated, by flipping micromagnets at the junctions. The implementation of the Poisson-Schrödinger equation to model MZM in the 3D geometry of a single nanowire could be integrated with 3D stray field profiles rather than simplified one-dimensional profiles integrated over the nanowire cross-section, which will provide information on the effect of the non-uniformity of the field combined with the effect of electrostatic confinement and disorder.

To summarize, nanoscale control over the field magnitude and direction is advantageous, particularly for advanced geometries where some nanowire segments may run perpendicu-

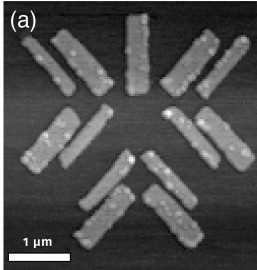 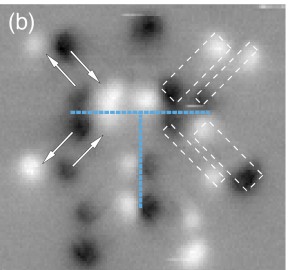

Figure 7: (a) Atomic force microscopy (AFM) of a T-junction setup with three Dragonfly magnet configurations. Magnetic film thickness is 20 nm. (b) Magnetic force microscopy (MFM) data on a different T-Junction of the same design. Arrows indicate magnetization direction, white dashed lines are example of magnet dimensions. Blue dashed line is where the Majorana nanowire is envisioned. The magnetic history was a magnetizing field from 100 mT to -16 mT to 0 mT applied at 45 degree direction, then 100 mT to -16 mT to 0 mT at 135 degrees.

lar to others. Micromagnets could offer novel ways to control Majorana-based devices, for instance by switching magnetization in order to reposition or couple MZMs.

**Acknowledgements and funding.** We thank S. Meynell and D. Yang for helpful discussions. Work in Pittsburgh (numerics and nanofabrication) supported by the Department of Energy DE-SC-0019274. A.B.J. acknowledges the support of the NSF Quantum Foundry through Q-AMASE-i program award DMR-1906325.

**Code and data availability.** Example MuMax3 scripts, numerical results data, Majorana simulation Mathematica notebooks are available on Zenodo [33], or GitHub via [https://github.com/frolovgroup](https://github.com/frolovgroup).

**Author Contributions.** M.J. and E.J. performed MuMax3 simulations. M.J., E.J. and J.S. performed Majorana simulations. Y.J. fabricated micromagnet devices. W.W. and A.B.J. performed MFM/AFM imaging. M.J. and S.F. wrote the manuscript with input from co-authors.

# A  Supplementary Information

## A.1  Background on Majorana nanowire model

MZMs are non-Abelian in nature and are their own antiparticle. They appear as topological, zero-energy states localized at the ends of a 1D system. They were first theoretically investigated in condensed matter systems by Kitaev in one-dimensional, spinless (p-wave), topological superconductors [34]. However, p-wave superconductors have yet to be realized experimentally, and, therefore, models that are experimentally accessible have been utilized in recent times. The prospect for MZMs that this work is utilizes is using semiconductor-nanowire, superconductor hybrid systems under external magnetic fields [1,2]. These hybrid systems utilize the nanowire's inherently strong spin-orbit coupling; proximity-induced s-wave superconductivity via contact with conventional superconductors; and the magnetic field to act as effective p-wave superconductors.

The Kitaev chain model is described by this Hamiltonian [34],

$$\hat{H}_p = -\mu \sum_i \left[ c_i^\dagger c_i \right] - \frac{1}{2} \sum_i \left[ t c_i^\dagger c_{i+1} + \Delta e^{i\phi} c_i c_{i+1} + H.c. \right], \tag{A.1}$$

where $c_i$ destroys a spinless fermion at site i, $c_i^\dagger$ creates a spinless fermion, $\mu$ is the chemical potential, $t > 0$ is the site hopping energy and $\Delta e^{i\phi}$ is the p-wave superconducting pairing

term. When this model is written in terms of Majorana fermion operators $\gamma_1$ and $\gamma_2$ Kitaev showed that this describes a system with Majorana fermions that have on site and nearest neighbour interactions. The key insight was that in the topological regime, defined by $|\mu| < t$, unpaired Majorana fermions are left at the ends of the 1D chain which could be potentially detected and manipulated.

To be able to effectively describe p-wave superconductors in a real system the semiconductor nanowire model is used, with Hamiltonian:

$$\hat{H}_{NW} = \begin{array}{l} \left( \frac{p^2}{2m^*} - \mu + \alpha_R \left( \boldsymbol{\sigma} \times \mathbf{p} \right) \cdot \hat{\mathbf{z}} \right) \tau_z \\ + V_z \left( \cos\theta\, \sigma_x + \sin\theta\, \sigma_y \right) + \Delta_s \tau_x \sigma_y, \end{array} \tag{A.2}$$

which acts on the Nambu spinor basis $\hat{\psi}_x = \left( c_{\uparrow x}, c_{\downarrow x}, c_{\uparrow x}^\dagger, -c_{\downarrow x}^\dagger \right)$. The momentum $\mathbf{p}$ points in the direction of a section of nanowire, $\Delta_s$ is the s-wave superconducting pairing term, $V_z = \frac{1}{2} g_{eff} \mu_B B$ is the Zeeman energy, $\theta$ is the angle of the field relative to the positive x-axis, $\tau$ and $\sigma$ are Pauli matrices representing the particle-hole and spin spaces respectively, and $\alpha_R$ is the Rashba spin orbit coupling term. The SOC term has $\cdot\hat{\mathbf{z}}$ as this is the direction of the internal electric field which induces the Rashba SOC, this direction arises because of the crystal asymmetry from the nanowire growing on the substrate, and the nanowire coating can affect this as well [35]. These parameters are taken constant because the nanowire is considered to be made of a homogeneous material. To be in the topological phase the following condition must be met

$$V_z > \sqrt{\mu^2 + \Delta_s^2}. \tag{A.3}$$

When this system is deep in the topological phase it was shown that this reduces to a p-wave topological system described by Eq.(A.1). To simulate the model in a finite one-dimensional system the continuum model is discretized on a 1D lattice in the electronic state basis, giving Eq. (A.4).

## A.2 Further Reading

For basic introductory review of Majorana nanowire topic consider reading [36]. A perspective giving an explanation on Rashba spin-orbit coupling one can read [35]. Other relevant work on nanowires that deal with areas such as surface effects, effects of the contacts, electric field considerations, and try to comprehensively implement the model [26,37]. For work that utilizes the Poisson Schrödinger equation to model MZMs in the 3D geometry of a single nanowire [26,37,38], where the use of 3D field profiles could give a batter picture of systems in future studies. Further work on implementing micromagnets for use in Majorana based devices [29]. There are several different proposed braiding schemes using nanowire-network junction devices [26,39], with options such as using local chemical potential changes [32] or flux gates [40].

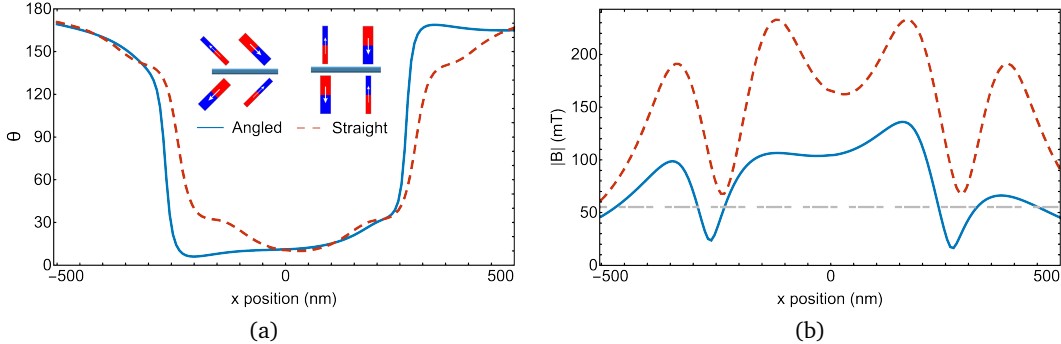

Figure 8: This compares the a rotated and un-rotated Dragonfly configuration. The solid line is the rotated set-up and the dashed line is the straight set-up. Rotating the magnets makes them easier to magnetize through hysteresis. This figure is for a 1000 nm nanowire segment.

## A.3   Majorana Model Used

The implemented one-dimensional semiconductor-superconductor hybrid model is given by a discretized Hamiltonian,

$$
\begin{aligned}
\hat{H} = \ & \sum_n \left[ (-\mu_n + \varepsilon_0) |n\rangle \langle n| \, \tau_z \otimes \sigma_0 \right. \\
& + t(a)(|n+1\rangle \langle n| + H.c.) \, \tau_z \otimes \sigma_0 \\
& + i\alpha(a)(|n+1\rangle \langle n| - H.c.) \, \tau_z \otimes (\boldsymbol{\sigma} \times \mathbf{x}) \cdot \hat{\mathbf{z}} \\
& + \tau_z \otimes V_{z,n} \left( \cos\theta_n \sigma_x + \sin\theta_n \sigma_y \right) |n\rangle \langle n| \\
& \left. - \Delta_s \tau_x \otimes \sigma_x |n\rangle \langle n| \right],
\end{aligned}
\tag{A.4}
$$

where

$$
t(a) = \frac{\hbar^2}{2m^* a^2} \,, \qquad \alpha(a) = \frac{\alpha_R}{2a} \,.
\tag{A.5}
$$

The meaning and values used for the parameters are as follows: $n$ labels the lattice site, $a$ is the lattice constant and $\varepsilon_0 = \cos(ka)$ with $k$ the crystal momentum and this is an offset to bring the Fermi energy to the bottom of the band. To deal with nanowires in different directions $\mathbf{x}$ is a unit vector pointing along a particular section of nanowire. The s-wave superconducting pairing term is $\Delta_s = 0.08$ meV, the Rashba spin orbit coupling term $\alpha_R = 0.2$ eVÅ, effective electron mass $m^* = 0.04 m_e$, and $g_{eff} = 50$ such as in InSb wires. Lastly, the Zeeman energy is $V_{z,n} = \frac{1}{2} g_{eff} \mu_B B_n$, with the magnetic field, $B_n$, and relative angle of the field to the x-axis, $\theta_n$, being both site dependent and in general have a complicated structure due to the use of the micromagnets, a notable difference to many other works. The simulated magnetic field from MuMax3 is inputted into the 1D nanowire-superconductor model via $V_{z,n}$, and the spectrum and eigenstates calculated via exact diagonalization. The effects of the non-uniform structure of the Zeeman energy were utilized and investigated by constructing longer devices with many micromagnets, and a T-junction set-up where perpendicular wires had different field directions.

Interpreting Majorana results. To investigate viability of the proposed devices, the MZMs signatures of state polarization and zero-energy pinning are required. A single electronic Majorana state, with energy $E_0$, can be written in terms of the two Majorana wavefunctions via

$\gamma_1 = c + c^\dagger$ and $\gamma_2 = -i\left(c - c^\dagger\right)$ and, due to the topological nature of MZMs, polarization of the two MZM will be apparent, seen as the localisation of two Majoranas at opposite ends of a single topological region of nanowire. In short length systems the MZMs can be overlapping, and also could have a complicated structure due to the variation of the magnetic field or nanowire length. The lowest eigenstates of the system are single quantum states that are pairs of MZMs. The first energy/excited state, denoted $E_1$ if there is a single pair of MZMs present, should have weight predominately in the bulk, with little overlap with the MZMs. The second signature is the near-zero energy of the MZM state, with good energy separation from the first energy state.

## A.4  Micromagnetic Simulation Details

**MuMax3**  [25] is a GPU-accelerated micromagnetic simulation program that uses finite difference discretization methods, accurate sized magnetic domains within a single magnet for micro- to nano- scale system simulations, and was implemented to obtain the magnetic field used in the hybrid nanowire model. The micromagnetic structure is first constructed for each set-up in MuMax3, the requisite magnetization states achieved through hysteresis, if possible, and then the stray field is averaged over the hexagonal nanowire cross-section (such as for InSb wires) with diagonal width of 100 nm, typical of experiments. Upon averaging, an effective one-dimensional magnetic field profile is obtained. Note that the magnetic field can be in principle calculated for an arbitrary coordinate, meaning that the MuMax calculation does not need to encode a fixed nanowire length. This one-dimensional field profile is then used in the Majorana model to calculate several lowest energy energies and wavefunctions. At this step a finite nanowire length is chosen. The code used for the calculations can be found on Zenodo [33].

**Hysteresis.**   To obtain the required magnetization state through hysteresis the micromagnets start with many randomly orientated domains and then an external magnetizing field is applied and the system was relaxed at regular time intervals while ramping up and down the field. It was found that angling the micromagnets at 45 degrees made obtaining the desired magnetization directions easier, as magnets on the opposite side of the nanowires were then perpendicular to each other. This allows the micromagnets to be closer together, however angling the magnets reduces the magnetic field, see Fig. 8, so these effects have to be balanced to obtain the strongest magnetic field possible. The hysteresis process takes into account the material of magnets, which include saturation magnetization, anisotropy constants and exchange stiffness. One limitation present is the material of the nanowire, leads and other components were not taken into account in the simulation, but the main magnetic effects are presumed to come from the magnets themselves.

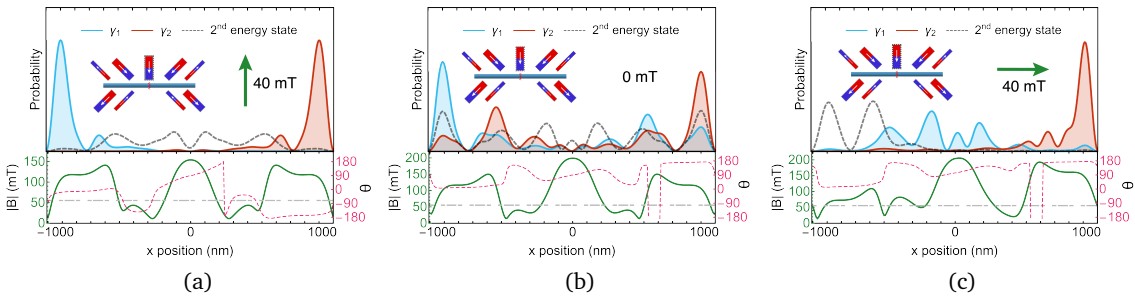

Figure 9: Comparing the double Dragonfly with different external fields. (a) 40 mT in the positive y direction (b) no field and (c) 40 mT in the positive x direction.

**The magnetic material parameters**   of Cobalt are used due to several factors, first is that the constructed micromagnet devices are thought to be mainly composed of CoFe. Additionally, Cobalt is a common material used in the fabrication of micromagnets, and it allows for a relatively high saturation magnetization, and thus a stronger stray field, while also allowing for a reasonable coercive field due to Cobalt's exchange stiffness and cubic anisotropy constant. This allows for an appropriate demagnetization/stray field in the nanowire region, while also reliably obtaining the magnetization direction required for the setup, and additionally allowing for a small external field to be applied without accidentally flipping the magnets. The parameters used are a saturation magnetisation of $1.44 \times 10^6$ A/m, exchange stiffness $3.0 \times 10^{-11}$ J/m, anisotropy constant $K_1$ $4.5 \times 10^5$, anisotropy constant $K_2$ $1.5 \times 10^5$ and a Landau-Lifshitz damping constant of 0.01.

Micromagnets of two different widths are reliable and sufficient enough for hysteresis and field production purposes, the dimensions of all magnets in this work are either that of the thin magnet, with dimensions $130 \times 1000 \times 100$ nm, or the thick magnet with dimensions $250 \times 1000 \times 100$ nm, as the different sized magnets flip at different coercive fields. The nanowire used in the MuMax simulations has a hexagonal cross-section with a long diagonal length of 100 nm.

## A.5   Supplementary Results

### A.5.1   Dragonfly

For the Dragonfly setup, the positions and dimensions of the set-up are as follows. The micromagnets used were two different widths, with thin magnet dimensions $130 \times 1000 \times 100$ nm or thick magnet with dimensions $250 \times 1000 \times 100$ nm, so the different sized magnets flip at different coercive fields. The magnets are rotated at 45 degrees to aid the reliability of magnetization directions. The nanowire has a hexagonal cross-section, with a long diagonal length of 100 nm, and length of 700 nm so it spanned the region of parallel field from the magnets. The centers of the micromagnets are displaced from the middle of the nanowire to the left and right by 250 nm, with magnets of the different widths opposite each other. The y-displacement of the micromagnets from the nanowire is chosen such that the nearest corner of each magnet is 40 nm from the nanowire. These positions are one workable configuration, and changing the positions, dimensions or magnet material parameters will change the magnetic field profile, but this is the optimal set-up found.

Due to the magnetic field being calculated through simulated hysteresis the magnets are not perfect single domains, in particular at the edges of the micromagnets the boundary causes non-uniformities. However, the net effect of this is small, but it explains why some of the state's energies and positions are not perfectly symmetric. Lastly, we note the difficulty to obtain a field larger than $\sim$ 150 mT in the current set-up, additionally the superconducting gap is not necessarily fixed experimentally, which will change the field requirements.

### A.5.2   Double Dragonfly without extra magnet

This double Dragonfly device (inset Fig. 10(a)), is different from the one shown in the main text as it only combines two individual Dragonfly set-ups (without the middle 9th magnet). The magnetic field has a large region of low magnitude field in the middle section, see bottom panel of Fig. 10(b), and there is an external field of 40mT applied in positive y-direction. The eigenstate profiles and energy spectrum suggest there are two pairs of uncoupled MZM on each side of the wire.

### A.5.3  Double Dragonfly with extra Magnet

Full dimensions of the double Dragonfly with the extra magnet are given in Fig.11. A gate is turned on in the middle region of the nanowire (red zig-zag line), uncoupling the ends of the nanowire and causing two pairs of MZMs to appear. This is implemented as a large potential barrier spanning around 10 sites, and would be implemented experimentally by pinching off the nanowire. These two pairs of MZMs live in the parallel region of field, similar to the single Dragonfly set-up, with energies of $E_0/\Delta = 3.1 \times 10^{-2}$ and $E_1/\Delta = 3.2 \times 10^{-2}$, and the first excited state at a larger energy of $E_2/\Delta = 8.9 \times 10^{-1}$.

The states in the double Dragonfly device under different external magnetic fields are shown in 9. Panel (a) is the same figure in the main text with an external field in the y-direction which aids coupling. Compare to panel (b) without the external field where the MZM states are less clear with less polarization and larger energies. Panel (c) shows how applying a Bx field with the double Dragonfly set-up gives preference to one side over the other. This is what would happen if a double Dragonfly was used in the leg section of the T-junction.

### A.5.4  T-junction magnetic fields

The magnetic field profile input into the nanowire model was made by taking the simulated central three single Dragonfly set-ups (with the middle magnet) from MuMax3, and then the top field profile was elongated by attaching two flipped copies of the top field profile and and connecting these on the ends. This had to done as it wasn't possible to accurately simulate all the magnets in the final T-junction set-up, however due to the modular nature of the Dragonfly set-ups this is reasonable. The magnetic field in the top nanowire rotates smoothly across the length of the nanowire (see Fig. 14a), and the whole nanowire is one, mostly continuous topological region. The magnetic field in the perpendicular section of nanowire is similar to that of the single Dragonfly set-up, with a parallel region of around 700 nm, note this is very uniformly pointing up (positive y) due to magnets on opposite sides of the nanowire having the same widths, see Fig. 14b. The applied external magnetic field, of 40 mT, was added in the positive y-direction to facilitate maximum coupling.

### A.5.5  Field uniformity

To show the uniformity of the field near and inside the nanowire the following is considered, with all data on Zenodo. The single micromagnet in Fig. 15 shows the field is only highly non-uniform very near the micromagnet poles, and drops off rapidly away from the micromagnet. Note that the field is very weak at the magnet sides, hence why a multiple magnet design is required to obtain strong enough stray fields. The most inhomogeneous region is therefore where the magnets come closest to the nanowire Fig. 16 which shows the field magnitude in the area between two micromagnets in a 5 nm thick layer. In the region of the nanowire (marked by red lines) the field variation is significant near the tips of the magnets, but the field is much more uniform away from the tips on the left. Fig. 17 plots the standard deviation of the field magnitude over the nanowire cross-section, note this is for a finer resolution than Fig 3. The standard deviation is relatively low compared to the mean for the nanowire segment in between the four magnets, away from the magnet tips. Over the majority of the nanowire the standard deviation is less than 10% of the mean. Slices inside the hexagonal nanowire region show the same. Full 3D data are available online.

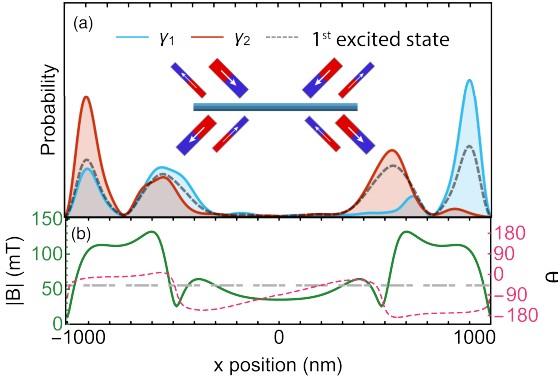

Figure 10: (a) Probability distributions for two lowest energy states, this system shows two overlapping and degenerate states with no clear Majorana polarization and no higher energy state being gapped out. The energies of both states $E_1/\Delta = 3.9 \times 10^{-2}$ and $E_2/\Delta = 4.6 \times 10^{-2}$. (b) Field profile along wire. External field is 40mT in positive y-direction.

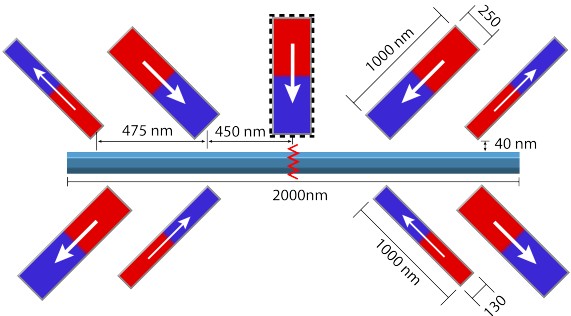

Figure 11: The double Dragonfly setup with the magnet in the middle and a potential gate, dimensions given. Toggling this gate allows the left and right side of the wire to be coupled or uncoupled.

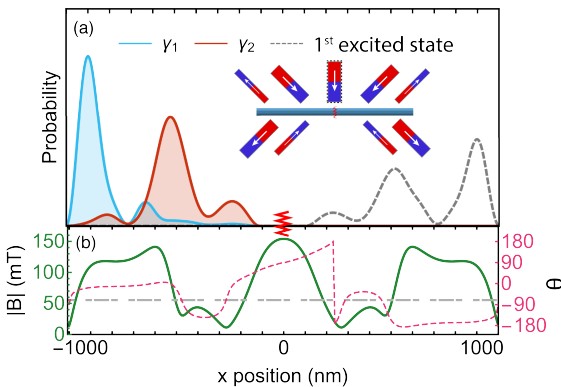

Figure 12: (a) Probability distributions for two lowest energy states for double Dragonfly setup with the 9th magnet and the gate closed (inset). The probability distributions shows two separate pairs of MZMs in each section of the wire, the left is shown in the Majorana basis with energy $E_1/\Delta = 3.1 \times 10^{-2}$ (red and cyan), the other pair in the right section is in the electronic state basis with energy $E_2/\Delta = 3.1 \times 10^{-2}$ (black- dashed line). The third state is well separated at $E_3/\Delta = 8.9 \times 10^{-1}$. (b) Field profile along wire. External field is 40mT in positive y-direction.

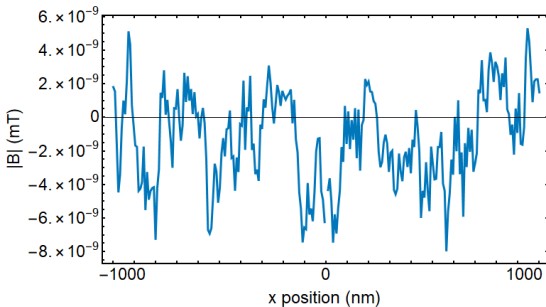

Figure 13: $B_z$ field component along the wire for the double Dragonfly with the middle ninth magnet. The $B_z$ data for other set-ups available on Zenodo [33].

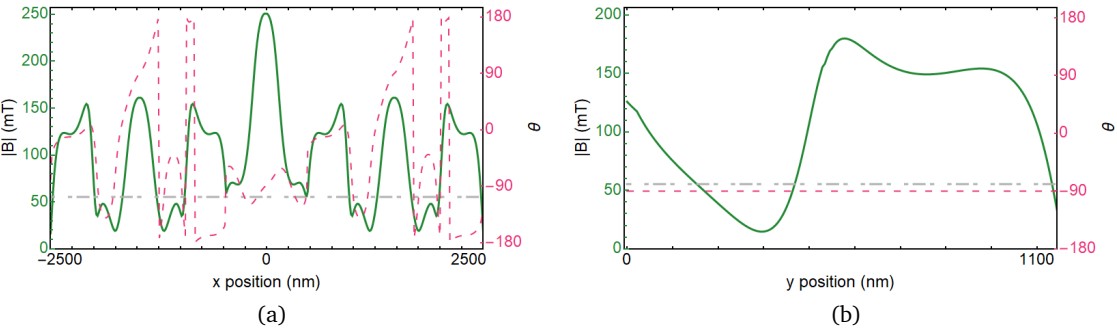

(a)                (b)

Figure 14: (a) Magnetic field of top wire for T-Junction. (b) Magnetic field in the T-Junction leg (vertical segment of nanowire). There is an external field of 40mT applied in positive y-direction.

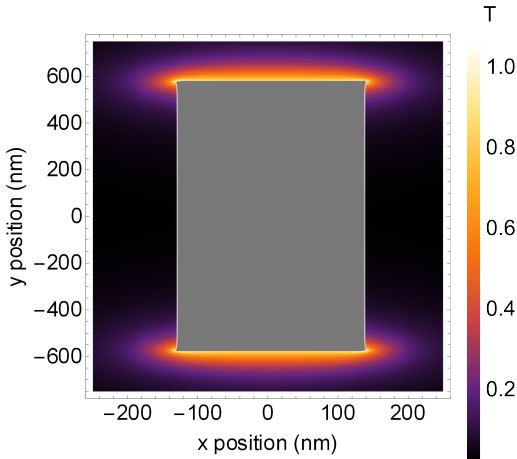

Figure 15: A heat map for magnetic field magnitude for a micromagnet of dimensions 230X1000X100 nm. This is shown for a 2D X-Y plane slice through the middle of the micromagnet (50nm), the magnet is shown as grey. Note the field is very weak away from the micromagnet ends.

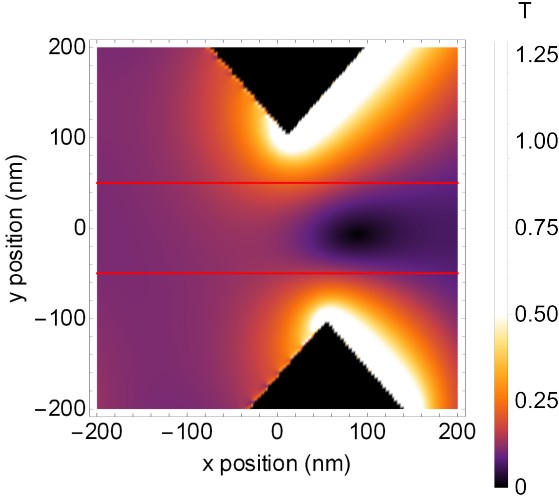

Figure 16: A X-Y plane heat map for magnetic field magnitude in a small region near the nanowire, the magnet areas have been set to 0 field (black triangular regions). This is for a 5 nm thick slice in the middle where the nanowire is (marked by red lines). A Mathematica script is available online to look at all the different X-Y slices at different z-positions.

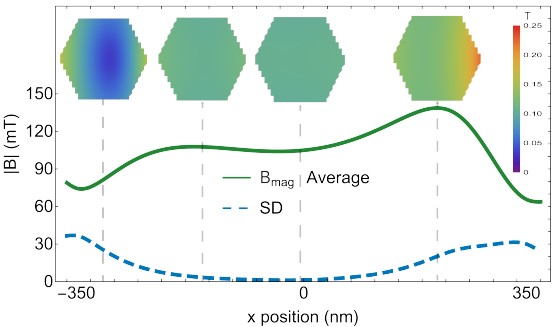

Figure 17: Shows the magnetic field magnitude's standard deviation (dashed) across the hexagonal cross-section and mean value over the nanowire region for the single Dragonfly set-up. The insets are field magnitude cross sections of the nanowire at different positions. Note the mean value is the same data that is plotted in Fig. 3 (b). A Mathematica script is available on Zenodo to look at all Y-Z slices along the nanowire.

Figure 18: Magnetic force microscopy (MFM) data on a 24 T-junction set-ups using three Dragonfly's. The magnet dimensions are the same as in Fig. 1 but the distance between them differs. Top left: Atomic force microscopy (AFM) of one T-junction setup with three dragonfly magnet configurations. Magnetic film height 20 nm. Dashed ovals point out magnetizations favorable for generating MZM. The magnetic history was a magnetizing field from 100 mT to -16 mT to 0 mT applied at 45 degree direction, then 100 mT to -16 mT to 0 mT at 135 degrees.

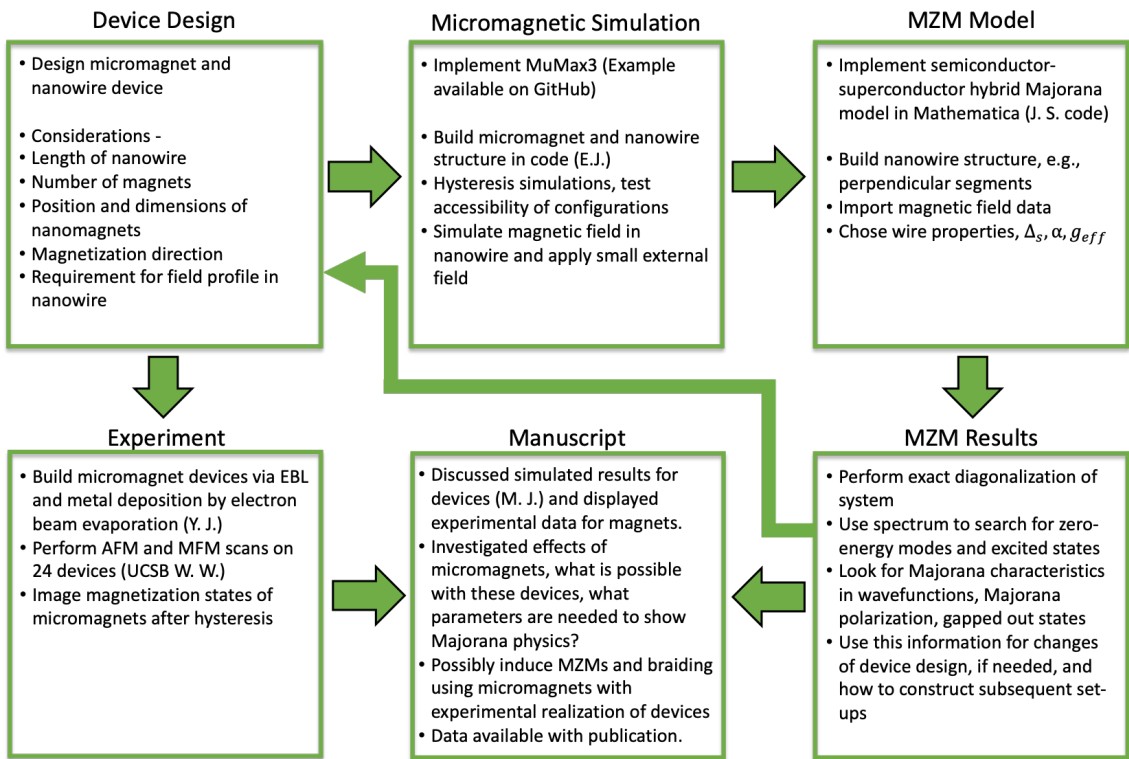

Figure 19: **Study design, Volume and Duration of study.** The bulk of the this work was carried out over nine months to find an appropriate micromagnet and nanowire set-up that demonstrated MZMs in the hybrid model. Approximately 5 different single Dragonfly configurations were tested and simulated, which was then used to construct the double Dragonfly and the T-junction. There were around 10 different designs tested for the double Dragonfly and T-junction. The experimental data was collected on a single chip where 24 devices had AFM and MFM measurements taken after an hysteresis cycle was applied to magnetize the configurations.

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
