# Peer review of "Integrating micromagnets and hybrid nanowires for topological quantum computing"

_SciPost Physics, doi:SciPost Phys. 11, 090 (2021)_

## Round 1 · Referee Report · Anonymous (Referee 2) · 2021-5-18

Report

The authors propose and study a new way to induce locally a magnetic field using micromagnets in order to allow the emergence of Majorana zero modes in a single nanowire as well as in more complex devices such as T-junctions. Numerical stray field simulations are presented along with numerical simulations of Majorana wavefunctions and energy spectrum at one dimension. First experimental realization of an inhomogeneous stray field required for a T-junction and produced by micromagnets is shown, pointing both the feasibility and the limitations of the proposal.

The original idea presented in this work constitutes an interesting and new experimental technique that should be considered in the race for performing topological quantum computing based on Majorana qubits. Overall, the results and analysis support the main conclusions and claims and the novelty of the approach presented here is of interest for the field of Majorana nanowires. I recommend it therefore for publication. Nevertheless, the paper suffers from unclarities. More precisely, the following issues should be addressed in a revised version prior to any publication.

Requested changes

1) The advantage of the method is not clearly pointed out in the introduction. It is only shortly mentioned in the text and the reader has to make some effort to gather together the different advantages scattered all over the text. According to the present work, this method allows to avoid any external magnetic fields to induce MZM and to locally induce stray fields that can be oriented differently in the samples. Therefore, it opens the way to the measurement of complicated devices like T-junctions braiding devices. This would be also the case of nanowire being covered by an insulating ferromagnetic layer. Nevertheless, the authors implied in the text that the stray field induced in the nanowire with the present method should much more homogeneous. Another advantage might the greater freedom offered to address and/or manipulate individually the MZM.

Some issues are not mentioned or are insufficiently discussed in the present version of the manuscript: 2) In order to generate a MZM, the nanowire needs to be covered by a superconducting thin film. Due to the Meissner effect, such a superconducting layer will locally influence the configuration of the stray field. 3) The effect of the presence of a disorder that is definitively present in any nanowire should be discussed and how the MZM are sensitive to such a static disorder. 4) The authors assume for their numerical simulations that the stray field is constant over the all cross section of the nanowire. Some arguments to support this claim are missing in the text. How homogeneous is the field in the y-direction? How would be the emergence of MZM sensitive to any inhomogeneous stray field in the y direction? 5) It looks like the experimental realization of the micromagnets have a different aspect ratio than the one considered in numerical simulations. How will the aspect ratio impact the configuration of the stray field? 6) One can notice some tiny asymmetry between right and left (red and green) in the double Dragonfly as well as in the T-junction when the problem is supposed to be mirror symmetric. This can be seen in the wavefunction as well as in the energy levels of the system. Is there any explanation for this asymmetry? 7) In the double Dragonfly configuration, not only the total length is responsible for the reduction of the overlap of the MZM but also the decay of the wavefunction that appears to be stronger by comparing figures 3.a and 4.a. Some sentences are not clear and should be rephrased or developed: 8) “Nevertheless, left and right MZM form a pair due to field rotation provided by the central magnet.” A non-expert reader might appreciate to have more details there. 9) The term “activated” is confusing. You mean that the wire is “pinched off”.

Some less important remarks: 10) There is no reference to S4, S6, S9 in the main text and S7 is only mentioned in the supplementary informations. 11) You mention “Twenty-four T-junctions with three Dragonflies each are imaged, and six of 76 total Dragonflies”: it corresponds to 72 configurations (24 x 3), doesn’t it? 12) There is some color mistake in the figure 4 as well as in figures S1, S2 and S3: the small magnet below the NW has the wrong color (blue <-> red) whereas the arrow indicates the right magnetization.

  • validity: good
  • significance: good
  • originality: high
  • clarity: ok
  • formatting: reasonable
  • grammar: -

Author:  Sergey Frolov  on 2021-07-15  [id 1573]

(in reply to Report 2 on 2021-05-18)
Category:
answer to question
correction
validation or rederivation

1) The advantage of the method is not clearly pointed out in the introduction. It is only shortly mentioned in the text and the reader has to make some effort to gather together the different advantages scattered all over the text. According to the present work, this method allows to avoid any external magnetic fields to induce MZM and to locally induce stray fields that can be oriented differently in the samples. Therefore, it opens the way to the measurement of complicated devices like T-junctions braiding devices. This would be also the case of nanowire being covered by an insulating ferromagnetic layer. Nevertheless, the authors implied in the text that the stray field induced in the nanowire with the present method should much more homogeneous. Another advantage might the greater freedom offered to address and/or manipulate individually the MZM.

Added statement in introduction.

Based on arXiv:2104.01623 we do not believe that structures with ferromagnetic insulator shells are capable of inducing MZM in semiconductor nanowires via the exchange effect. However, those structures also do produce stray magnetic fields which influence the nanowire, though the profiles of those fields would not be suitable for the generation of MZM.

Some issues are not mentioned or are insufficiently discussed in the present version of the manuscript: 2) In order to generate a MZM, the nanowire needs to be covered by a superconducting thin film. Due to the Meissner effect, such a superconducting layer will locally influence the configuration of the stray field.

3) The effect of the presence of a disorder that is definitively present in any nanowire should be discussed and how the MZM are sensitive to such a static disorder.

Added a statements in the section ‘Limitations and Conclusions’

4) The authors assume for their numerical simulations that the stray field is constant over the all cross section of the nanowire. Some arguments to support this claim are missing in the text. How homogeneous is the field in the y-direction? How would be the emergence of MZM sensitive to any inhomogeneous stray field in the y direction?

Copying response to the first referee who asked the same question.

We added a new section in the supplementary (field uniformity) with the data requested and the standard deviation of the field magnitude along the nanowire. We provide 3D MuMax data as a Mathematica notebook that can be plotted in various forms. The standard deviation of magnetic field across the cross-section is at the 5% level within the 700-nm segment used in the Majorana simulation for Figure 3. In conclusion and limitations we discuss how the 3D model, including Poisson-Schrodinger and 3D MuMax field profile can be integrated to do a more quantitative analysis of our concept.

5) It looks like the experimental realization of the micromagnets have a different aspect ratio than the one considered in numerical simulations. How will the aspect ratio impact the configuration of the stray field?

The dimensions of each micromagnet are the same in the imaging experiment and in the numerical model. However, the distance between the magnets is different. The micromagnets are weakly influenced by each other at these distances so the effect of the distance on the orientations of the magnets is small. We did not investigate the stray field profile in imaging but only the magnet orientations, due to the technique resolution.

6) One can notice some tiny asymmetry between right and left (red and green) in the double Dragonfly as well as in the T-junction when the problem is supposed to be mirror symmetric. This can be seen in the wavefunction as well as in the energy levels of the system. Is there any explanation for this asymmetry?

Any asymmetry is likely an effect of the relaxation of the domains in a single micromagnets in MuMax3, in particular the edges of the magnets aren’t perfect as the edge is a boundary for the individual magnetic domains that make up a single magnet.

Added statement in Supplementary Results - Dragonfly to explain this detail.

7) In the double Dragonfly configuration, not only the total length is responsible for the reduction of the overlap of the MZM but also the decay of the wavefunction that appears to be stronger by comparing figures 3.a and 4.a.

In the double dragonfly the two MZM in the middle of the device hybridize strongly and this reduces their overlap with the end MZMs. We clarified and corrected the discussion.

Some sentences are not clear and should be rephrased or developed: 8) “Nevertheless, left and right MZM form a pair due to field rotation provided by the central magnet.” A non-expert reader might appreciate to have more details there.

Explained that MZM generated by opposite fields should naively not couple.

9) The term “activated” is confusing. You mean that the wire is “pinched off”.

We call this ‘pinch-off’ now.

Some less important remarks: 10) There is no reference to S4, S6, S9 in the main text and S7 is only mentioned in the supplementary information.

We reordered and re-referenced several of these figures in the supplementary. There are some figures that are only relevant for details in the supplementary, such as the MuMax details for the (reordered S1).

11) You mention “Twenty-four T-junctions with three Dragonflies each are imaged, and six of 76 total Dragonflies”: it corresponds to 72 configurations (24 x 3), doesn’t it?

Correct! Fixed

12) There is some color mistake in the figure 4 as well as in figures S1, S2 and S3: the small magnet below the NW has the wrong color (blue <-> red) whereas the arrow indicates the right magnetization.

Fixed

---

## Round 1 · Referee Report · Kristof Moors (Referee 1) · 2021-5-18

Strengths

1)
The authors propose an original architecture that attempts to meet both basic requirements (for realizing Majorana zero modes) and practical requirements (for setting up the micromagnets).
2)
The article contains both numerical simulations and an experimental proof-of-concept.
3)
The source code and scripts of the simulations and plots are publicly available in a git repository.

Weaknesses

1)
While being an original proposal, I am not convinced that the proposal is particularly promising for realizing a scabable Majorana architecture. The initialization of the micromagnets in the proof-of-concept is not yet completely successful and the nanowires still have to be incorporated, as well as additional hardware for initialization, manipulation, and readout (e.g., gates).
2)
The presentation of the results could be improved.

Report

The authors propose an original micromagnet architecture (with a so-called Dragonfly as building block) to realize scalable networks for Majorana zero modes without requirement of a (strong) external magnetic field. While the proposed architecture and experimental proof-of-concept is not fully convincing, I think that this work is worth publishing in SciPost Physics and could inspire further research in this direction of combining Majorana nanowire in network structures with micromagnets or, more generally, nonuniform magnetization profiles. I do find that the presentation of the results are hard to follow in some places and the manuscript could definitely benefit from some improvements, so my verdict is minor revision. The problems with the presentation are listed in detail below, in the section for requested changes.

Requested changes

1) I would like to see a statement on the validity of the 1D model with magnetization profiled averaged in the transverse direction. Based on Fig. 1, it seems that there could be a strong variation in the transverse direction, especially where the micromagnet ends are close to the nanowire, and this might invalidate the results obtained with the 1D model to some extent. In this context, it would also be interesting to see the vector field with a bit more detail inside and near the nanowire region (a zoom-in of Fig. 1 with nanowire region visible).

2) The method section on the micromagnetic simulations in the main text is very brief. Some basic information on the simulation approach and the parametrization of the micromagents could be added there. Or, at the very least, a reference to the relevant supplementary section must be added.

3) The position of the nanowire and extent of the topological region should be clearly indicated on Figs. 2-4, S1, S3-S5, S7. And another suggestion for improvement: replace Fig. 2 or 3b by a vector field (now they are exact copies). In general, I think that a vector field is much easier to read than plotting the profile $\theta(x)$. Also, the introduction of $\theta$ in Fig. 2 is not the most intuitive when considering the reference frame depicted in Fig. 1.

3) Is there a particular reason why the T-junction setup does not contain a double Dragonfly in the bottom leg? I would expect that it would further improve the Majorana polarization. I would like to see some statement of this design choice.

4) It would be much easier to parse the list of energies in Fig. 6 when they are presented in a table and when the energies of the red, green, and blue colored states in the subfigures are indicated on an axis. (minor comment: it is not clearly stated that these values are for pairs of states)

5) On Eq. (4) in the Supplementary Information: $\epsilon_0$ is not defined and the momentum operator $\mathbf{p}$ should not appear anymore.

6) The color coding of one of the micromagnets on the bottom right of the double Dragonfly (on Figs. 2, S1, S2, S3) is incorrect, and also of one of the wave functions on Fig. 6d (purple instead of red).

7) Some figures in the Supplementary are not referred to in the text and some don't appear in the right order.

8) typos: -'Schroedinger' on p. 1 - 'micromagentic' on p. 2 - 'mircomagnets' on p. 4 - 'cite i' (-> site $i$) on p. 7 - 'disretized' on p. 7 - 'Shrödinger' on p. 4,7 - 'exited' (-> excited) on p. 9 - 'x position' (-> y position) on Fig. S5b

  • validity: good
  • significance: ok
  • originality: high
  • clarity: ok
  • formatting: good
  • grammar: -

Author:  Sergey Frolov  on 2021-07-15  [id 1572]

(in reply to Report 1 by Kristof Moors on 2021-05-18)
Category:
answer to question
reply to objection
correction

>1)I would like to see a statement on the validity of the 1D model with magnetization profiled averaged in the transverse direction. Based on Fig. 1, it seems that there could be a strong variation in the transverse direction, especially where the micromagnet ends are close to the nanowire, and this might invalidate the results obtained with the 1D model to some extent. In this context, it would also be interesting to see the vector field with a bit more detail inside and near the nanowire region (a zoom-in of Fig. 1 with nanowire region visible).

We added a new section in the supplementary (field uniformity) with the data requested and the standard deviation of the field magnitude along the nanowire. We provide 3D MuMax data as a Mathematica notebook that can be plotted in various forms. The standard deviation of magnetic field across the cross-section is at the 5% level within the 700-nm segment used in the Majorana simulation for Figure 3.
In conclusion and limitations we discuss how the 3D model, including Poisson-Schrodinger and 3D MuMax field profile can be integrated to do a more quantitative analysis of our concept.

>2)The method section on the micromagnetic simulations in the main text is very brief. Some basic information on the simulation approach and the parametrization of the micromagents could be added there. Or, at the very least, a reference to the relevant supplementary section must be added.

We added a sentence referring to the supplementary materials to the brief methods section. We would like to keep that section brief.

>3)The position of the nanowire and extent of the topological region should be clearly indicated on Figs. 2-4, S1, S3-S5, S7.

The position of the nanowire for Figures 2 and 3 is shown in Figure 1. For Figure 4 it is shown in Figure S3. The way the field profile is generated for the T-junction is explained in the section ‘T-junction braiding device’.
We did not evaluate the extent of the topological region. Rather, we solved the Schroedinger equation with the Hamiltonian given by Equation (2) in the entire nanowire as it is shown in the figures. We show directly the lowest energy wavefunctions that came out of this calculation, which are decomposed in the left-right Majorana basis. If ‘topological region’ is defined as a region in which there exists a non-zero left-right Majorana separation, then the entire nanowire is topological.

>- 3.5)And another suggestion for improvement: replace Fig. 2 or 3b by a vector field (now they are exact copies). In general, I think that a vector field is much easier to read than plotting the profile θ(x). Also, the introduction of θ in Fig. 2 is not the most intuitive when considering the reference frame depicted in Fig. 1.

The choice of using field angle to visualize the field was made because the angle is quantitatively important, as deviation of field from the nanowire axis suppresses the topological gap. This sentence was clarified and 2 relevant references were added (Phys. Rev. B, 89:245405, Jun 2014., Phys. Rev. B, 90:115429, Sep 2014) “The relatively parallel field profile is required for MZMs formation, since such field is mostly orthogonal to the effective spin-orbit field, Bso [1, 2, 27, 28], hence why we use angle relative to the nanowire when presenting the stray field profile.”.

The use of the 2 similar figures is didactic, first introducing the magnetic field configuration and then showing the wavefunctions. When we tried to plot fields in vector representation people had a hard time understanding the data, but original data available with the paper can be plotted in this form.
To help with the reference frame we state “relative to negative x-axis (inset)” in the Fig2 caption now.

>3) Is there a particular reason why the T-junction setup does not contain a double Dragonfly in the bottom leg? I would expect that it would further improve the Majorana polarization. I would like to see some statement of this design choice.

The reason is that we apply By=40mT to make the field better aligned with the horizontal segment of the T-junction. We do the same for single and double dragonfly. However, for the leg segment this field is along the segment. And in that case it does not lead to significantly better Majorana separation for a double dragonfly vs a single dragonfly. We have added a figure in the supplementary that shows what happens to the double dragonfly for no external field, for y field and for x field.

>4)It would be much easier to parse the list of energies in Fig. 6 when they are presented in a table and when the energies of the red, green, and blue colored states in the subfigures are indicated on an axis. (minor comment: it is not clearly stated that these values are for pairs of states)

We added a table in the caption area to hold the data on energies and colours of the states.
We also clarified that the energies are for pairs of Majorana states. The lowest eigenstates of the system are single quantum states that are pairs of Majorana.

>5) On Eq. (4) in the Supplementary Information: ϵ0 is not defined and the momentum operator p should not appear anymore.

Fixed p appearing twice. epsilon_0 is an offset to bring the energy near the bottom of the band, added this.

>6)The color coding of one of the micromagnets on the bottom right of the double Dragonfly (on Figs. 2, S1, S2, S3) is incorrect, and also of one of the wave functions on Fig. 6d (purple instead of red).

The magnet’s colours are now correct.
For Fig. 6d the purple is because there is some of the blue state in the same position as red, added statement in text.

>7) Some figures in the Supplementary are not referred to in the text and some don't appear in the right order.

Changed to more appropriate order and made sure the figures are appropriately referenced in the text

>8)typos:

all fixed

---

## Round 2 · Referee Report · Kristof Moors (Referee 1) · 2021-7-29

Report

I still have a few minor points that I would like to see addressed before accepting the paper for publication.

  • One point that I would like to clarify was my earlier comment 3 on the position of the nanowire on the figures and the indication of the topological regime. It is confusing that the actual extent of the nanowire in Figs. 2 and S1 is not indicated. The x position extends over 900 nm and 1000 nm, respectively, whereas the nanowire in Fig. 1 is only 700 nm long. It should be specified where the wire is considered to be positioned in these plots (it might be obvious that the wire sits between -350 and 350, but it is not explicitly stated). I find it especially confusing as the profile is obtained by averaging over the hexagonal cross section of a nanowire, also at positions beyond the extent of the actual nanowire. In the other figures, there is no mismatch between (horizontal) plot range and the extent of the nanowire structure, so I would say that there is no ambiguity there.

With the indication of the topological region, I was referring to the extent of the wire where the green curve stays above the gray dashed line (indicating the minimal uniform field for entering the topological regime), in Figs. 2 and 3, for example. Hence, one could consider this interval to be the effective length of the topological wire section, which is different from the total length of the wire.

  • Another point is the appearance of the momentum operator in Eq. (4). I was not referring to the reappearance in the text, but rather in the formula itself. After discretizing the continuum model of Eq. (2), the momentum operator should disappear from the Hamiltonian, as momentum gets represented by hopping between neighboring lattice sites in the lattice model (just like the momentum squared in the first kinetic term does not appear any longer and gets replaced by hopping terms).

  • Then, there is the addition of Fig. S10, with a statement on the standard deviation being only 5% in the main text (I suppose that this is the overall SD over the total wire length). I think that this is a bit misleading, as there are positions where the cross section is much less uniform, especially near the wire ends. Although, I am not too worried about that, actually, because the profile is the most uniform in the topological section of the wire. Perhaps this is worth stressing rather than just mentioning the overall standard deviation of 5%.

Otherwise, everything has been dealt with appropriately and I accept the paper for publication when these minor points have been addressed.

Requested changes

  • address ambiguity regarding nanowire position in Figs. 2 and S1

  • fix Hamiltonian in Eq. (4)

  • clarify statement on field uniformity and meaning of standard deviation in main text

  • Small typo in the conclusion: "Disorder is real nanowire devices ..." -> "Disorder in real nanowire devices ..."

  • validity: -
  • significance: -
  • originality: -
  • clarity: -
  • formatting: -
  • grammar: -

Author:  Sergey Frolov  on 2021-10-01  [id 1796]

(in reply to Report 1 by Kristof Moors on 2021-07-29)

  • address ambiguity regarding nanowire position in Figs. 2 and S1

We added comments in the text to improve clarity.

We point out that there is no ambiguity here, because the micromagnetic simulation is independent of the nanowire length, so the effective 1D magnetic field profile can be calculated without assuming a fixed nanowire length. This is what is done in Figures 2 and S1 where we wanted to show what the 1D profile looks like for a longer range of coordinate.

  • fix Hamiltonian in Eq. (4)

This is fixed. Thanks!

  • Small typo in the conclusion: "Disorder is real nanowire devices ..." -> "Disorder in real nanowire devices ..."

Fixed. Thanks!

  • clarify statement on field uniformity and meaning of standard deviation in main text

Clarified that the standard deviation quoted is not for field variations along the nanowire, but for field variations within the cross-section. Full data from which this statement can be understood and evaluated are provided in supplementary and on Zenodo.

---

## Round 2 · Referee Report · Anonymous (Referee 2) · 2021-8-6

Report

The authors answered my questions and comments appropriately. I would therefore recommend the paper for publication.

---

## Round 2 · Author Response

Requested changes >1)I would like to see a statement on the validity of the 1D model with magnetization profiled averaged in the transverse direction. Based on Fig. 1, it seems that there could be a strong variation in the transverse direction, especially where the micromagnet ends are close to the nanowire, and this might invalidate the results obtained with the 1D model to some extent. In this context, it would also be interesting to see the vector field with a bit more detail inside and near the nanowire region (a zoom-in of Fig. 1 with nanowire region visible).

We added a new section in the supplementary (field uniformity) with the data requested and the standard deviation of the field magnitude along the nanowire. We provide 3D MuMax data as a Mathematica notebook that can be plotted in various forms. The standard deviation of magnetic field across the cross-section is at the 5% level within the 700-nm segment used in the Majorana simulation for Figure 3. In conclusion and limitations we discuss how the 3D model, including Poisson-Schrodinger and 3D MuMax field profile can be integrated to do a more quantitative analysis of our concept.

>2)The method section on the micromagnetic simulations in the main text is very brief. Some basic information on the simulation approach and the parametrization of the micromagents could be added there. Or, at the very least, a reference to the relevant supplementary section must be added.

We added a sentence referring to the supplementary materials to the brief methods section. We would like to keep that section brief.

>3)The position of the nanowire and extent of the topological region should be clearly indicated on Figs. 2-4, S1, S3-S5, S7. The position of the nanowire for Figures 2 and 3 is shown in Figure 1. For Figure 4 it is shown in Figure S3. The way the field profile is generated for the T-junction is explained in the section ‘T-junction braiding device’.

We did not evaluate the extent of the topological region. Rather, we solved the Schroedinger equation with the Hamiltonian given by Equation (2) in the entire nanowire as it is shown in the figures. We show directly the lowest energy wavefunctions that came out of this calculation, which are decomposed in the left-right Majorana basis. If ‘topological region’ is defined as a region in which there exists a non-zero left-right Majorana separation, then the entire nanowire is topological.

>- 3.5)And another suggestion for improvement: replace Fig. 2 or 3b by a vector field (now they are exact copies). In general, I think that a vector field is much easier to read than plotting the profile θ(x). Also, the introduction of θ in Fig. 2 is not the most intuitive when considering the reference frame depicted in Fig. 1.

The choice of using field angle to visualize the field was made because the angle is quantitatively important, as deviation of field from the nanowire axis suppresses the topological gap. This sentence was clarified and 2 relevant references were added (Phys. Rev. B, 89:245405, Jun 2014., Phys. Rev. B, 90:115429, Sep 2014) “The relatively parallel field profile is required for MZMs formation, since such field is mostly orthogonal to the effective spin-orbit field, Bso [1, 2, 27, 28], hence why we use angle relative to the nanowire when presenting the stray field profile.”. The use of the 2 similar figures is didactic, first introducing the magnetic field configuration and then showing the wavefunctions. When we tried to plot fields in vector representation people had a hard time understanding the data, but original data available with the paper can be plotted in this form. To help with the reference frame we state “relative to negative x-axis (inset)” in the Fig2 caption now.

>3) Is there a particular reason why the T-junction setup does not contain a double Dragonfly in the bottom leg? I would expect that it would further improve the Majorana polarization. I would like to see some statement of this design choice. The reason is that we apply By=40mT to make the field better aligned with the horizontal segment of the T-junction. We do the same for single and double dragonfly. However, for the leg segment this field is along the segment. And in that case it does not lead to significantly better Majorana separation for a double dragonfly vs a single dragonfly. We have added a figure in the supplementary that shows what happens to the double dragonfly for no external field, for y field and for x field.

>4)It would be much easier to parse the list of energies in Fig. 6 when they are presented in a table and when the energies of the red, green, and blue colored states in the subfigures are indicated on an axis. (minor comment: it is not clearly stated that these values are for pairs of states)

We added a table in the caption area to hold the data on energies and colours of the states. We also clarified that the energies are for pairs of Majorana states. The lowest eigenstates of the system are single quantum states that are pairs of Majorana.

>5) On Eq. (4) in the Supplementary Information: ϵ0 is not defined and the momentum operator p should not appear anymore.

Fixed p appearing twice. epsilon_0 is an offset to bring the energy near the bottom of the band, added this.

>6)The color coding of one of the micromagnets on the bottom right of the double Dragonfly (on Figs. 2, S1, S2, S3) is incorrect, and also of one of the wave functions on Fig. 6d (purple instead of red).

The magnet’s colours are now correct. For Fig. 6d the purple is because there is some of the blue state in the same position as red, added statement in text. >7) Some figures in the Supplementary are not referred to in the text and some don't appear in the right order.

Changed to more appropriate order and made sure the figures are appropriately referenced in the text

>8)typos:

all fixed

Anonymous Report 2 on 2021-5-18

>1) The advantage of the method is not clearly pointed out in the introduction. It is only shortly mentioned in the text and the reader has to make some effort to gather together the different advantages scattered all over the text. According to the present work, this method allows to avoid any external magnetic fields to induce MZM and to locally induce stray fields that can be oriented differently in the samples. Therefore, it opens the way to the measurement of complicated devices like T-junctions braiding devices. This would be also the case of nanowire being covered by an insulating ferromagnetic layer. Nevertheless, the authors implied in the text that the stray field induced in the nanowire with the present method should much more homogeneous. Another advantage might the greater freedom offered to address and/or manipulate individually the MZM.

Added statement in introduction.

Based on arXiv:2104.01623 we do not believe that structures with ferromagnetic insulator shells are capable of inducing MZM in semiconductor nanowires via the exchange effect. However, those structures also do produce stray magnetic fields which influence the nanowire, though the profiles of those fields would not be suitable for the generation of MZM.

>Some issues are not mentioned or are insufficiently discussed in the present version of the manuscript: 2) In order to generate a MZM, the nanowire needs to be covered by a superconducting thin film. Due to the Meissner effect, such a superconducting layer will locally influence the configuration of the stray field. >3) The effect of the presence of a disorder that is definitively present in any nanowire should be discussed and how the MZM are sensitive to such a static disorder.

Added a statements in the section ‘Limitations and Conclusions’

>4) The authors assume for their numerical simulations that the stray field is constant over the all cross section of the nanowire. Some arguments to support this claim are missing in the text. How homogeneous is the field in the y-direction? How would be the emergence of MZM sensitive to any inhomogeneous stray field in the y direction?

Copying response to the first referee who asked the same question.

We added a new section in the supplementary (field uniformity) with the data requested and the standard deviation of the field magnitude along the nanowire. We provide 3D MuMax data as a Mathematica notebook that can be plotted in various forms. The standard deviation of magnetic field across the cross-section is at the 5% level within the 700-nm segment used in the Majorana simulation for Figure 3. In conclusion and limitations we discuss how the 3D model, including Poisson-Schrodinger and 3D MuMax field profile can be integrated to do a more quantitative analysis of our concept.

>5) It looks like the experimental realization of the micromagnets have a different aspect ratio than the one considered in numerical simulations. How will the aspect ratio impact the configuration of the stray field?

The dimensions of each micromagnet are the same in the imaging experiment and in the numerical model. However, the distance between the magnets is different. The micromagnets are weakly influenced by each other at these distances so the effect of the distance on the orientations of the magnets is small. We did not investigate the stray field profile in imaging but only the magnet orientations, due to the technique resolution.

>6) One can notice some tiny asymmetry between right and left (red and green) in the double Dragonfly as well as in the T-junction when the problem is supposed to be mirror symmetric. This can be seen in the wavefunction as well as in the energy levels of the system. Is there any explanation for this asymmetry?

Any asymmetry is likely an effect of the relaxation of the domains in a single micromagnets in MuMax3, in particular the edges of the magnets aren’t perfect as the edge is a boundary for the individual magnetic domains that make up a single magnet.

Added statement in Supplementary Results - Dragonfly to explain this detail.

>7) In the double Dragonfly configuration, not only the total length is responsible for the reduction of the overlap of the MZM but also the decay of the wavefunction that appears to be stronger by comparing figures 3.a and 4.a.

In the double dragonfly the two MZM in the middle of the device hybridize strongly and this reduces their overlap with the end MZMs. We clarified and corrected the discussion.

>Some sentences are not clear and should be rephrased or developed: 8) “Nevertheless, left and right MZM form a pair due to field rotation provided by the central magnet.” A non-expert reader might appreciate to have more details there.

Explained that MZM generated by opposite fields should naively not couple.

>9) The term “activated” is confusing. You mean that the wire is “pinched off”.

We call this ‘pinch-off’ now.

>Some less important remarks: >10) There is no reference to S4, S6, S9 in the main text and S7 is only mentioned in the supplementary information.

We reordered and re-referenced several of these figures in the supplementary. There are some figures that are only relevant for details in the supplementary, such as the MuMax details for the (reordered S1).

>11) You mention “Twenty-four T-junctions with three Dragonflies each are imaged, and six of 76 total Dragonflies”: it corresponds to 72 configurations (24 x 3), doesn’t it?

Correct! Fixed

>12) There is some color mistake in the figure 4 as well as in figures S1, S2 and S3: the small magnet below the NW has the wrong color (blue <-> red) whereas the arrow indicates the right magnetization.

Fixed

---

## Round 2 · List of Changes

Changes are visible in blue here: https://arxiv.org/src/2104.05130v2/anc/manuscript_with_edits_highlighted.pdf

---

## Editorial Decision

published